# A Universal Error Measure for Input Predictions Applied to Online Graph Problems

**Giulia Bernardini**
University of Trieste, Italy
giulia.bernardini@units.it

**Alexander Lindermayr**
University of Bremen, Germany
linderal@uni-bremen.de

**Alberto Marchetti-Spaccamela**
La Sapienza University of Rome, Italy
alberto@diag.uniroma1.it

**Nicole Megow**
University of Bremen, Germany
nicole.megow@uni-bremen.de

**Leen Stougie**
CWI and Vrije Universiteit
Amsterdam, The Netherlands
Leen.Stougie@cwi.nl

**Michelle Sweering**
CWI, Amsterdam, The Netherlands
Michelle.Sweering@cwi.nl

## Abstract

We introduce a novel measure for quantifying the error in input predictions. The error is based on a minimum-cost hyperedge cover in a suitably defined hypergraph and provides a general template which we apply to online graph problems. The measure captures errors due to absent predicted requests as well as unpredicted actual requests; hence, predicted and actual inputs can be of arbitrary size. We achieve refined performance guarantees for previously studied network design problems in the online-list model, such as Steiner tree and facility location. Further, we initiate the study of learning-augmented algorithms for online routing problems, such as the online traveling salesperson problem and the online dial-a-ride problem, where (transportation) requests arrive over time (online-time model). We provide a general algorithmic framework and we give error-dependent performance bounds that improve upon known worst-case barriers, when given accurate predictions, at the cost of slightly increased worst-case bounds when given predictions of arbitrary quality.

## 1 Introduction

We develop a novel measure for quantifying the error in input predictions and apply it to derive error-dependent performance guarantees of algorithms for online metric graph problems. Online graph problems are among the most fundamental online optimization problems, where an initially unknown input is revealed incrementally. The two main paradigms for the incremental information release are the *online-time* model and the *online-list* model. In the online-time model, initially unknown requests are revealed over time and can be served any time, whereas in the online-list model, requests are revealed one-by-one and must be served immediately before the next request appears. In this work, we address specifically the following online routing and network design problems.

**Online-time routing problems** In the classical *Online Traveling Salesperson Problem* (OLTSP) and *Online Dial-a-Ride Problem* (OLDARP), a server can move at unit speed in a given metric space. Transportation requests appear online over time, each defining a start and end point in the metric space (in the TSP both points are equal). The task is to determine a tour to serve all requests (in

36th Conference on Neural Information Processing Systems (NeurIPS 2022).

any order) by moving to the corresponding start and end points. The objective is to minimize the makespan, i.e., the time point when all requests have been served and the server is back in the origin. These problems are well-studied [7, 8, 17, 18, 24], as well as other related variants [31, 32, 36].

**Online-list network design problems** In the *Online Steiner Tree Problem*, requests are terminal nodes that are revealed one-by-one in a given metric space (typically represented as a complete edge-weighted graph) and must be connected to a fixed root by selecting edges via other (Steiner) nodes. In the closely related *Online Steiner Forest Problem*, a request is composed of two nodes which have to be connected by the selected set of edges. In both problems, the objective is to minimize the total cost of selected edges. In the more general *Online Facility Location Problem*, a facility can be opened at every vertex at a certain one-time cost at any time, and arriving client vertices are connected upon arrival to the closest open facility at the cost of the shortest path to it. The goal is to minimize the opening and connection cost. These problems are very well-studied [2–4, 9, 13, 15, 20, 21, 23, 25, 26, 30, 39, 45, 46].

The performance of online algorithms is typically assessed by worst-case analysis. An algorithm is called $\rho$-*competitive* if it computes, for any input instance, a solution with objective value within a multiplicative factor $\rho$ of the optimal value that can be computed when knowing the full instance upfront. The *competitive ratio* of an algorithm is the smallest factor $\rho$ for which it is $\rho$-competitive. For the above problems (nearly) tight bounds on the competitive ratio are known. For OLTSP and OLDARP, there have been shown best possible 2-competitive algorithms [7, 8]. For the online network design problems, the existence of $O(1)$-competitive algorithms has been ruled out [25, 30, 39] and algorithms with (tight) (poly-)logarithmic upper bounds have been shown [15, 25, 30, 39, 45].

The assumption in online optimization of not having any prior knowledge about future requests seems overly pessimistic. In particular, given the success of machine-learning methods and data-driven applications, one may expect to have access to predictions about future requests. However, simply trusting such predictions might lead to very poor solutions, as these predictions come with no quality guarantee. The recent vibrant line of research initiated in [37, 38] aims at incorporating such error-prone predictions into online algorithms, to go beyond worst-case barriers. The goal are *learning-augmented algorithms* with a performance that is close to that of an optimal offline algorithm when given accurate predictions (called *consistency*) and, at the same time, never being (much) worse than that of a best known algorithm without access to predictions (called *robustness*). Further, the performance of an algorithm shall degrade in a controlled way with increasing prediction error.

In this paper, we consider an *input predictions* model, i.e., there is given a set $\hat{R}$ of predictions for the actual online input $R$ of a problem. We do not make any assumption on the quality of the prediction or on its size. In particular, $\hat{R}$ might be substantially larger or smaller than $R$.

Defining an appropriate error measure is a crucial task in this line of research. There is no common agreement (yet) in the literature on what constitutes a good error measure. The philosophy behind our error measure is the following. Any learning-augmented algorithm needs to trust the predictions to some extent, as otherwise no improvement upon an online algorithm is possible. The error should then be able to sensitively bound how much any (reasonable) algorithm pays for erroneous predictions. Roughly, our error measure achieves this by approximating the extra cost that an algorithm trusting the predictions has when it serves the true instance; very informally, this is $\mathrm{OPT}_{\mathrm{trust}\,\hat{R}}(R) - \mathrm{OPT}(R)$. Several natural measures (for graph problems) have been proposed, such as the number of erroneous predictions [47], the $\ell_1$-norm (e.g., distances between predicted and real points), or more involved perfect matching-based errors [12]. Although we cannot expect that a single error measure is appropriate for all problems, we propose a universal template based on the cost of a hyperedge cover in a bipartite hypergraph that is constructed in a problem-specific way.

## 1.1 Our contributions

**Cover error for input predictions** Here we sketch the main idea of our error measure, which will be made precise in Section 2. We separately cover the errors incurred by *unexpected actual requests*, $R \setminus \hat{R}$, and *absent predicted requests*, $\hat{R} \setminus R$, as these pose a potential threat to an algorithm which trusts $\hat{R}$. For each of the two error types we consider a suitable weighted bipartite hypergraph with erroneous requests on the left side and define an error measure combining the costs of minimum hyperedge covers of the left side of each hypergraph. Let us concentrate on errors due to unexpected actual requests, being the nodes on the left side, with the predicted requests as the nodes on the right side. Each hyperedge links a single node on the right side with a subset of the nodes on the left side

which it *covers*. Its contribution to the overall error, i.e., its cost in the hyperedge cover problem to cover all left side nodes, is related to the optimal cost for the subinstance induced by its nodes. E.g. in OLTSP this cost is the value of an optimal tour for some unexpected requests (left) when starting from some predicted request (right), which can be seen as a minimum detour that needs to be made from the predicted request to serve the unexpected requests. Bounding the number of left side requests in the hyperedges by $k$ yields a hierarchical family $\{\Lambda_k\}_{k=1}^{\infty}$ of errors, with higher values of $k$ giving errors that reflect more precisely the cost due to trusting wrong predictions.

The cover error fulfills several useful properties. First, it provides a framework that may apply to various problems by assigning appropriate costs to the hyperedges. E.g. for the online-time model it allows to integrate in a very precise way actual and predicted release dates, as we demonstrate in Section 3. This is a feature which previous metric graph errors seem to miss [12, 47], because they rely on counting incorrect predictions or disallow asymmetric cost functions. Our error also naturally supports different sizes of $R$ and $\hat{R}$, reflecting the input sequence length being unknown in almost all online optimization problems in the literature. Although previously studied error measures [12, 47] do support this in theory, we will show in Section 2 that they fail to detect good predictions in certain scenarios, which results in imprecise weak performance bounds. In contrast, the cover error guarantees an almost optimal performance of the same algorithms in these cases. We therefore hope that the cover error will be useful for better analyzing existing learning-augmented algorithms and other problems in the future.

**Algorithms with error-sensitive performance bounds** Our algorithmic results are twofold: we provide the first learning-augmented algorithms for online-time routing problems, and we give new error-dependencies for existing algorithms for online-list network design problems. The unifying element is that we achieve these by problem-dependent implementations of our new cover error. We first introduce a general framework for OLTSP and OLDARP, in which we delay the moment in which we start following the optimal predicted tour by a multiplicative trust factor $\alpha \in (0, 1)$. For robustness, before starting to follow the predictions any $\rho$-competitive online algorithm is executed in a black-box fashion. We prove an error-dependency w.r.t. the first cover error in the hierarchy $\Lambda_1$ (as properly defined in Section 2). We denote by $C^*$ the cost of an optimal tour on the actual requests $R$.

**Theorem 1.** OLTSP *and* OLDARP *admit learning-augmented algorithms that use a $\rho$-competitive algorithm as a subroutine and achieve a competitive ratio that is, for any $\alpha > 0$, bounded by* $\min\left\{(1 + \alpha)\left(1 + \frac{3 \cdot \Lambda_1}{C^*}\right), 1 + \rho + \frac{\rho}{\alpha}\right\}$.

Hence, sufficiently good predictions help to beat the classic lower bound of 2 [8] for OLTSP.

When using the algorithm of [7] as a subroutine, we can further refine our algorithm by carefully aligning the used waiting strategies and prove an improved robustness guarantee.

**Theorem 2.** OLTSP *and* OLDARP *admit a learning-augmented algorithm that uses the 2-competitive algorithm of [7] as a subroutine and achieves a competitive ratio that is, for any $\alpha > 0$, bounded by* $\min\left\{(1 + \alpha)\left(1 + \frac{3 \cdot \Lambda_1}{C^*}\right), 2 + \frac{2}{\alpha}\right\}$.

In general, online algorithms for OLTSP and OLDARP aim for tackling the uncertainty of the input rather than efficient running times, which is also the case for the above discussed results. Yet, we show in Section 3.2 that we can trade efficiency with slightly larger constant factors in the guarantees.

Further, we remark that simpler and tightened results are possible for restricted metric spaces: in Section 3.2 we provide improved bounds for OLTSP on the positive half of the real line. Here, a minimalistic prediction, a single value predicting the optimal makespan $C^*$, suffices to obtain an almost tight consistency-robustness tradeoff.

We complement these theoretical bounds with empirical results (see Section 5) on simulated real-world taxi instances in the city road network of Manhattan, which indicate the superior performance of our new algorithms compared to classic methods in both general and relevant restricted scenarios.

Further, we consider online-list graph problems and analyze the algorithmic framework provided by Azar, Panigrahi and Touitou [12] w.r.t. our new cover error. For each problem, we specify hyperedge cost functions which follow the same paradigm and prove new error-dependent bounds.

**Theorem 3.** *The algorithms in [12] for the online Steiner tree or online (capacitated) facility location problem incur, for any parameter $k \geq 1$, a cost of at most $\mathcal{O}(1) \cdot \mathrm{OPT} + \mathcal{O}(\log k) \cdot \Lambda_k$.*

**Theorem 4.** *The algorithm in [12] for the online Steiner forest problem incurs, for any parameter $k \geq 1$, a cost of at most $\mathcal{O}(1) \cdot \mathrm{OPT} + \mathcal{O}(k) \cdot \Lambda_k$.*

These bounds hold *simultaneously* for any $k$. The algorithm is still robust due to the robustness bound of $\mathcal{O}(\log|R|)$ provided by Azar et al. in [12]. On the technical side, we can exploit a technical lemma by [12] that allows us to split the analysis in two parts. The actual proofs for bounding the algorithm's cost by the cost of an optimal solution and the cover error are completely different.

For certain input scenarios we substantially strengthen the bounds on the competitive ratio provided by Azar et al. [12]. Indeed, their bound never improves over $\Omega(\log(\max\{|R|, |\hat{R}|\} - \min\{|R|, |\hat{R}|\}))$, which is not better than the best possible competitive ratio $\mathcal{O}(\log|R|)$ for classic online algorithms, if $|\hat{R}|$ and $|R|$ differ significantly. We show that there are input scenarios for which their error measure overestimates the actual error substantially and, thus, gives poor performance bounds while the algorithm performs actually well; more precisely, we prove a constant competitive ratio for the algorithm in [12] w.r.t. our error measure whereas the bound w.r.t. the previous measure is $\mathcal{O}(\log|R|)$.

### 1.2 Further related work

While untrusted predictions have been successfully integrated into online models for many different problems, none of the previous approaches and models seem to capture the complexity of combined routing and scheduling decisions. Related research includes work on scheduling [10, 11, 14, 29, 33, 34, 40, 42, 43], routing in metric spaces such as the k-server problem and more generally metrical task systems [5, 35], graph exploration [22] and online network design [1, 12, 47].

Further, there is hardly any work on integrating untrusted predictions into online problems in the online-time model. The only exception seems to be the work by Antoniadis et al. [6] on online speed scaling with a prediction model that includes release dates and deadlines. The nature of the speed-scaling problem, however, is very different from the routing problems we consider. Other works on non-clairvoyant scheduling with jobs arriving over time (minimizing flow time [10, 11], total completion time [34] and energy [14]) assume predictions on the job sizes or priorities; release dates are known in advance in [14], while [10, 11, 34] consider purely online problems w.r.t. job arrivals.

Very recently and independently of our work, two papers [27, 28] were announced that also study OLTSP in the learning-augmented setting. Hu et al. [28] consider OLTSP with different prediction models in general metric spaces. For arbitrary input predictions, their result has no error-dependency and a weaker consistency-robustness tradeoff compared to Theorem 2. Gouleakis et al. [27] exclusively study OLTSP on the real line. Assuming that the correct number of requests is known in advance, they study the power of predictions on the locations; their results are incomparable to ours.

### 1.3 Organization

We first introduce the cover error in Section 2, and then give our results for online-time routing problems and online-list network design problems in Section 3 resp. Section 4. Finally, we present empirical experiments in Section 5. Details on extensions, further experiments and missing proofs can be found in the full version [16] of this paper, which is also part of the supplemental material.

## 2 The cover error

Given an input prediction $\hat{R}$ and the actual input $R$, we design an error measure that *covers* every erroneously predicted item, i.e., all unexpected requests $R \setminus \hat{R}$ and all absent predicted requests $\hat{R} \setminus R$. As a concrete example think of OLTSP where a learning-augmented algorithm trusts (at least to a certain degree) the predicted requests in $\hat{R}$, and thus follows an optimal tour on $\hat{R}$. After serving a predicted request $(\hat{x}, \hat{r})$, it may serve some unexpected actual requests $R' \subseteq R \setminus \hat{R}$ that have already been released and are *relatively close* to $(\hat{x}, \hat{r})$ (both time- and location-wise). In our terminology, think of $(\hat{x}, \hat{r})$ covering $R'$, and observe that the cost for this cover shall naturally be equal to the optimal cost for serving $R'$ when starting in $\hat{x}$ at time $\hat{r}$. Conversely, the predicted requests that have not shown up and are nevertheless visited, $\hat{R} \setminus R$, should be covered by actual requests, to make up for the extra cost incurred by these superfluous visits.

We can arguably expect that any well-performing algorithm should be as least as good as serving all unexpected requests $R \setminus \hat{R}$ and all absent predicted requests $\hat{R} \setminus R$ in such partitions which can be covered by, respectively, predicted and actual requests in the cheapest possible way.

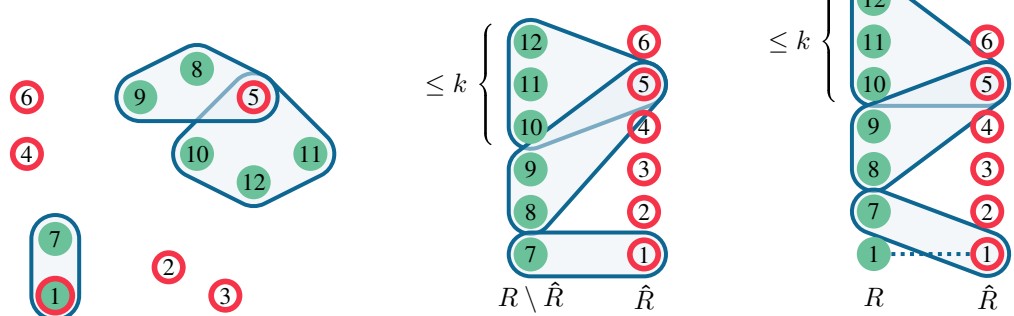

**(a)** Instance in metric space  **(b)** Min-cost $k$-hyperedge cover  **(c)** $\Gamma_k(R, \hat{R}) = \Gamma_k(R \setminus \hat{R}, \hat{R})$

**Figure 1:** Example for a metric instance and input prediction with a min-cost $k$-hyperedge cover of the set of unexpected requests $R \setminus \hat{R}$. The actual requests are filled green and the predicted requests are encircled red. The labels show which points in the metric space correspond to which nodes in the bipartite graphs.

We now embed this intuition into a precise definition. Let $A$ and $B$ be two sets of (possibly different) size and let $k \geq 1$. We define a bipartite hypergraph $G_k = (A \cup B, \mathcal{H})$ where $\mathcal{H}$ is the set of all hyperedges which have exactly one endpoint in $B$ and at most $k$ endpoints in $A$. A $k$-*hyperedge cover of $A$ by $B$* is a set of hyperedges $\mathcal{H}' \subseteq \mathcal{H}$ in $G_k$ such that every vertex in $A$ is incident to at least one hyperedge in $\mathcal{H}'$. If every hyperedge $h \in \mathcal{H}$ of $G_k$ has an associated cost $\gamma(h)$, a *minimum-cost $k$-hyperedge cover* $\mathcal{H}'$ is a $k$-hyperedge cover which minimizes the total hyperedge cost $\sum_{h' \in \mathcal{H}'} \gamma(h')$. We denote the value of a min-cost $k$-hyperedge cover of $A$ by $B$ by $\Gamma_k(A, B)$. Finally, the cover error, denoted by $\Lambda_k(R, \hat{R})$, is given by

$$\Lambda_k(R, \hat{R}) = \Gamma_\infty(\hat{R}, R) + \Gamma_k(R, \hat{R}).$$

Notice that we allow arbitrary large hyperedges ($k = \infty$) to cover predicted requests $\hat{R}$. We emphasize that all results also hold for a symmetric error definition $\Gamma_k(\hat{R}, R) + \Gamma_k(R, \hat{R})$, because $\Gamma_i(A, B) \geq \Gamma_{i+1}(A, B)$, for any $i$. Nevertheless we use this asymmetric definition to obtain a stronger bound when covering $\hat{R}$. Intuitively, this is possible because all predicted requests are known in advance (as opposed to the actual requests, which arrive online).

We simply write $\Lambda_k$ if $\hat{R}$ and $R$ are clear from the context. Since our error measure shall give value zero if $\hat{R} = R$, we require that the cost of every hyperedge $\{a\} \cup \{b\}$ for some $a \in A$ and $b \in B$ is equal to zero if $a = b$. Then, all vertices in $A \cap B$ can be covered trivially by $B$, and we conclude that $\Gamma_k(A \setminus B, B) = \Gamma_k(A, B)$. Figure 1 depicts an example of a $k$-hyperedge cover.

It remains to specify the cost $\gamma(A', b)$ of a hyperedge $A' \cup \{b\}$. Although we will give precise definitions separately for every concrete problem, all definitions follow a certain paradigm. That is, the cost $\gamma(A', b)$ shall be equal to *the value of an optimal solution for the subinstance induced by $A'$ with respect to $b$*. This anchoring requirement is the *single* detail which has to be specified for a concrete problem. Note that this matches our intuition discussed above for OLTSP.

**Comparison to other error measures** We compare the cover error to previously proposed error measures for the (undirected) online Steiner tree problem. Xu and Moseley [47] define a prediction error $\eta = \max\{|\hat{R}|, |R|\} - |\hat{R} \cap R|$, the number of erroneous requests, and prove that their algorithm is $\mathcal{O}(\log(\min\{|R|, \eta\}))$-competitive. Azar et al. [12] introduce the *metric error with outliers* $\lambda = (\Delta, D)$, where $D$ is the value of a min-cost perfect matching between two equally sized subsets of $R$ and $\hat{R}$, and $\Delta$ is the total number of unmatched points in $R$ and $\hat{R}$. They prove for their algorithms a multiplicative error dependency w.r.t. $\log(\min\{|R|, \Delta\})$ and an additive error dependency w.r.t. $D$.

We give a family of instances with $n$ actual requests and an input prediction for which the algorithms of Azar et al. [12] and Xu and Moseley [47] perform arguably well, but their error measures and analyses yield a bound of $\mathcal{O}(\log n) \cdot \text{OPT} + \mathcal{O}(\epsilon)$, which could be achieved even without predictions. For some $\epsilon > 0$, the instance is composed of one terminal request at $x_1$ and $n - 1$ requests in an $\epsilon$-ball around the Steiner point $x_2$, but no request is exactly on $x_2$. Both $x_1$ and $x_2$ are predicted. We can immediately observe that the number of erroneous requests [47] is $\eta = n - 1$. For the metric error with

outliers [12], note that any perfect matching is composed of at most two matches, therefore $\Delta \geq n-2$ and $D = \mathcal{O}(\epsilon)$. On the other hand, our cover error is bounded by $\Lambda_k \leq \Lambda_1 = \mathcal{O}(n \cdot \epsilon)$ for any $k$, because $x_2$ covers all requests in the $\epsilon$-ball around it. Then, Theorem 3 concludes that the algorithm of Azar et al. [12] is indeed constant competitive for this instance when $\epsilon \to 0$.

# 3 Online metric TSP with predictions

Let $M = (X, d)$ be a metric space, consisting of a set of points $X$, with origin $o \in X$ and a metric $d$. In the *Online Metric Traveling Salesperson Problem* (OLTSP), a set of unknown requests $R$ is released online over time. A request $(x, r)$ is composed of a point $x \in X$ and a release date $r \in \mathbb{R}_{\geq 0}$, i.e., the time at which the request becomes known and can be served. The task of an algorithm is to route a server, which is initially in the origin and moves at unit speed, through all requests back to the origin. The objective is to minimize the makespan, i.e., the total time required for this task.

The OLTSP *with predictions*, is an OLTSP in which we are given additionally an a priori prediction $\hat{R}$ on the set of requests. We assume that the server receives a signal when it is back at the origin after serving all the actual requests. Unlike in the classic OLTSP, this is important, as otherwise an algorithm might continue considering predicted requests, thus, ruling out any robustness.

To specify our cover error, we define the cost of a hyperedge $R' \cup \{(x', r')\}$ as the extra cost of serving erroneous (unexpected or predicted absent) requests $R'$ w.r.t. a request $x'$ (actual or predicted):

$$\gamma^{\mathrm{TSP}}(R', (x', r')) = \textit{optimal makespan for serving instance } R' \textit{ from origin } x' \textit{ and initial time } r'.$$

To get some intuition, consider $X = \mathbb{R}_{\geq 0}$ and an algorithm that does not move before time $t$ if there are no requests. It will receive an adversarial request $(t, t)$ and hence encounters a ratio of $\frac{3}{2}$ [18]. To overcome this when having (almost) accurate predictions, the server has to move towards (predicted) requests before they actually arrive. However, this pre-moving technique brings several challenges. If an algorithm moves the server without interruption to a predicted request at the beginning of the instance, an adversary would immediately spawn the single actual request at the origin, giving an unbounded robustness. If the server would directly move back, one can similarly argue that the consistency is at least $\frac{3}{2}$. Therefore, the key is to define a proper waiting strategy before moving towards predicted requests. We show that we can execute an arbitrary online algorithm while delaying pre-moving to gain information about the instance. This is very delicate, since too much delay clearly weakens the consistency, but too little delay gives weak robustness. We prove the following result.

**Theorem 5.** *Let $\alpha \in (0, 1/2)$ and let $\mathcal{A}$ be a $(1 + \alpha)$-consistent deterministic learning-augmented algorithm for* OLTSP. *Then, $\mathcal{A}$ can be $\beta$-robust only for $\beta \geq \frac{1}{\alpha} - 1$. This holds even on the half-line.*

Our final algorithm uses a hyperparameter $\alpha > 0$ to configure the waiting duration and thereby achieves a tight asymptotic consistency-robustness tradeoff. Intuitively, we can express our confidence in the prediction using $\alpha$ and get customized guarantees.

## 3.1 A general framework for OLTSP with predictions

Our strategy involves an initial delay phase in which we follow an arbitrary online algorithm, up to some predetermined time depending on the cost $\hat{C}$ of an optimal tour $\hat{T}$ of the predicted requests $\hat{R}$. After that, we start following $\hat{T}$, adjusting it whenever the actual requests deviate from the predictions. We call this greedy strategy PREDICTREPLAN (PREDREPLAN for short), due to the analogy with the classic REPLAN heuristic [7]. Let $p(t)$ be the server's location at time $t$.

---

**Algorithm 1** PREDREPLAN

Follow $\hat{T}$. Whenever an unexpected request $(x, r)$ is released, recompute and follow a fastest tour from $p(r)$ to the origin serving all unserved predicted requests as well as all the unserved unexpected requests. If the server receives an end signal in the origin, terminate.

---

While this algorithm might move towards predicted requests which are known to be absent to make the analysis clearer, a practical implementation ignores these and thereby only improve its performance.

We formally define the class of algorithms DELAYTRUST, that is parameterized by our trust parameter $\alpha > 0$, which scales the delay. Let $\mathcal{A}$ be any $\rho$-competitive online algorithm for OLTSP.

---

**Algorithm 2** DELAYTRUST

  i: Follow $\mathcal{A}$ as long as for time $t$ it holds $t \leq \alpha\hat{C} - d(p(t), o)$
  ii: Move the server to the origin
  iii: Follow the PREDREPLAN strategy until the end

---

We refer to the execution of each line as a *phase*. We now prove the main Theorem 1 for OLTSP, by showing for DELAYTRUST separately an error-dependent bound in Lemma 6 and a robustness bound in Lemma 7. Given an OLTSP instance, we denote by $C^*$ the makespan of an optimal tour $T^*$ serving all actual requests.

**Lemma 6.** DELAYTRUST *has a competitive ratio of at most* $(1 + \alpha)\left(1 + 3 \cdot \frac{\Lambda_1}{C^*}\right)$, *for any* $\alpha \geq 0$.

*Proof.* We first bound $\hat{C}$. Fix a min-cost $\infty$-hyperedge cover of $\hat{R}$ by $R$ and an optimal tour $T^*$ for $R$. For every hyperedge $\hat{R}' \cup \{(x, r)\}$ in the cover, we extend $T^*$ by adding the optimal offline OLTSP tour for $\hat{R}'$ which starts at $x$ at the time $t$ at which $T^*$ serves $x$. Note that, since $r \leq t$, the makespan of this subtour is bounded by the cost of $\hat{R}' \cup \{(x, r)\}$. Since every predicted request is covered by at least one hyperedge, the constructed tour serves $\hat{R}$ and we conclude that $\hat{C} \leq C^* + \Gamma_\infty(\hat{R}, R)$.

We now bound the makespan of the tour of the algorithm. If the algorithm terminates in Phases (i) or (ii), its makespan is at most $\alpha\hat{C} \leq \alpha \cdot (C^* + \Gamma_\infty(\hat{R}, R)) \leq (1 + \alpha) \cdot (C^* + \Lambda_1)$.

Otherwise, the algorithm reaches Phase (iii). There it first computes an optimal tour $\hat{T}$ of length at most $\hat{C}$ serving all unserved predicted requests. The makespan only increases when unexpected requests arrive. To this end, fix a min-cost 1-hyperedge cover of $R$ by $\hat{R}$ and a hyperedge $\{(x', r')\} \cup \{(\hat{x}, \hat{r})\}$ of this cover. We upper bound the additional cost due to $(x', r')$ by the cost of an excursion from the algorithm's current tour serving $(x', r')$. The algorithm might find a faster tour to serve all unserved requests and henceforth uses that. We distinguish two cases depending on the algorithm's remaining tour before request $(x', r')$ arrived. If $\hat{x}$ is not part of this tour, we consider an excursion which immediately deviates from $p(r')$ to serve $(x', r')$ and then returns to $p(r')$. By the triangle inequality, the length of this excursion is bounded by twice the distance between $p(r')$ and $\hat{x}$, plus the cost for optimally serving $(x', r')$ from $\hat{x}$ when starting at time $\hat{r}$. Due to our assumption, $(\hat{x}, \hat{r})$ must have already been served at some time $t$ with $r' \geq t \geq \hat{r}$. Thus, the algorithm's server is at most $r' - \hat{r}$ units away from $\hat{x}$ at time $r'$, and the total time incurred for this excursion is bounded by

$$2 \cdot (r' - \hat{r}) + \gamma^{\mathrm{TSP}}(\{(x', r')\}, (\hat{x}, \hat{r})) \leq 3 \cdot \gamma^{\mathrm{TSP}}(\{(x', r')\}, (\hat{x}, \hat{r})).$$

Note that the inequality is due to the fact that $(x', r')$ can only be served after its release date.

In the other case, the algorithm's server will visit $\hat{x}$ at some later point in time, especially at least once after time $\hat{r}$. We thus wait until the algorithm reaches $\hat{x}$ at some time $t \geq \hat{r}$, and then serve $(x', r')$ using at most $\gamma^{\mathrm{TSP}}(\{(x', r')\}, (\hat{x}, \hat{r}))$ additional time. See Figure 2 for an illustration of both cases.

Since every actual request is covered by one hyperedge, we conclude that Phase (iii) takes time at most $\hat{C} + 3 \cdot \Gamma_1(R, \hat{R})$. Adding the time for Phases (i) and (ii) gives a makespan of at most

$$(1+\alpha)\hat{C}+3\cdot\Gamma_1(R, \hat{R}) \leq (1+\alpha)\left(C^* + \Gamma_\infty(\hat{R}, R) + 3 \cdot \Gamma_1(R, \hat{R})\right) \leq (1+\alpha)\left(C^* + 3 \cdot \Lambda_1\right). \quad \square$$

**Lemma 7.** DELAYTRUST *has a competitive ratio of at most* $1 + \rho + \frac{\rho}{\alpha}$, *for any* $\alpha > 0$ *and any* $\rho$-competitive algorithm used in Phase (i).

*Proof.* If the algorithm terminates during Phase (i) or (ii), the competitive ratio is $\rho$. We are guaranteed to finish in one of these two phases if $\rho C^* \leq \alpha\hat{C}$.

If we terminate within Phase (iii), then $\hat{C} < \frac{\rho}{\alpha}C^*$. Once the last request has arrived at some time $r_{last} \leq C^*$, our tour stays fixed. We distinguish two cases. If the last request arrives before the end of Phase (ii), then the cost of our tour comprises of the cost for finishing Phase (ii), which is at

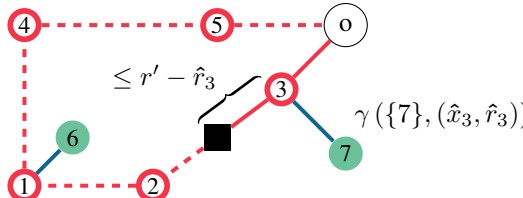

**(a)** The server (square) plans (dashed) and follows (solid) the predicted tour. Two hyperedges are completely released: {7} is covered by 3, which is already served, and {6} is covered by 1, which will be served in the future. Note that neither 1, 8 nor 9 have been released yet. Also notice that PREDREPLAN might find a faster tour.

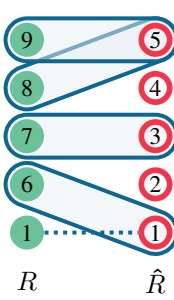

**(b)** Min-cost 1-hyperedge cover of $R$

**Figure 2**

most $\alpha\hat{C}$, and the cost of PREDREPLAN for following the predicted tour, including all unexpected yet unserved requests, which is at most $\hat{C} + C^*$. The total cost is thus bounded from above by

$$\alpha\hat{C} + \hat{C} + C^* \le \left(1 + (1+\alpha) \cdot \frac{\rho}{\alpha}\right) \cdot C^* = \left(1 + \rho + \frac{\rho}{\alpha}\right) \cdot C^*.$$

In the second case, the last request arrives in Phase (iii). In this case the cost after $r_{last}$ is the cost of following the predicted tour, adapted for incorporating the unexpected, yet unserved, requests. This is bounded above by the cost of returning to the origin, following the predicted tour $\hat{T}$, and finally following the optimal tour, $T^*$. Note that the cost of returning to the origin is at most $r_{last} - \alpha C^*$. Hence, we complete the proof by upper bounding the makespan, for any $\rho \ge 1$, by

$$r_{last} + r_{last} - \alpha\hat{C} + \hat{C} + C^* \le \left(3 + (1-\alpha) \cdot \frac{\rho}{\alpha}\right) C^* = \left(3 - \rho + \frac{\rho}{\alpha}\right) C^* \le \left(1 + \rho + \frac{\rho}{\alpha}\right) C^*. \quad \Box$$

### 3.2 Extensions and improvements

**An improved algorithm for** OLTSP **with predictions** A best possible online algorithm for OLTSP is SMARTSTART, which is 2-competitive [7]. Using this in Phase (i) of DELAYTRUST, Theorem 1 yields a robustness factor of at most $3 + \frac{2}{\alpha}$. We exploit SMARTSTART's waiting strategy to serve yet unserved requests and expedite Phase (iii) avoiding unnecessary waiting time, obtaining an algorithm, SMARTTRUST, with improved robustness factor $2 + \frac{2}{\alpha}$ as stated in Theorem 2.

**Algorithms with polynomial running time** Algorithms DELAYTRUST and SMARTTRUST require the computation of optimal TSP tours on subinstances. NP-hardness of TSP prohibits polynomial running time, unless P=NP. We provide performance guarantees for our learning-augmented algorithm framework when using polynomial-time $\nu$-approximation algorithms for solving TSP, which guarantee to find a TSP tour within a factor $\nu$ of the optimum.

We use a modified efficient PREDREPLAN strategy which uses a $\nu$-approximate solution instead of an optimal solution and further ensures that errors due to such approximations do not add up too much compared to our error budget $\Lambda_1$. Adjusting the proof of Theorem 1 yields the following result.

**Theorem 8.** *Given a $\nu$-approximation algorithm for metric TSP, the competitive ratio of the polynomial time* DELAYTRUST *using a polynomial time $\rho$-competitive online algorithm in Phase (i) is, for any $\alpha > 0$, bounded by* $\min\left\{(1+\nu)(1+\alpha)\left(1 + \frac{3 \cdot \Lambda_1}{2 \cdot C^*}\right), \left(\rho + (1+\nu)(1 + \frac{\rho}{\alpha})\right)\right\}.$

**Online metric Dial-a-Ride with predictions** The *Online Metric Dial-a-Ride Problem* (OLDARP) is a generalization of OLTSP where each request $(x^s, x^d, r)$ has a starting location $x^s$ and a destination $x^d$. To serve a request, the server must first visit $x^s$ at some time not earlier than $r$, and then $x^d$. We assume that the server can carry at most one request at the time and cannot store it after pickup.

We show that slight modifications of DELAYTRUST and SMARTTRUST yield Theorems 1 and 2 for this generalized setting. We define the cost function $\gamma^{\text{DaRP}}$ for the cover error:

$$\gamma^{\text{DaRP}}(R', (x^s, x^d, r)) = \min\{\gamma^{\text{TSP}}(R', (x^s, r)), \gamma^{\text{TSP}}(R', (x^d, r + d(x^s, x^d)))\} + D,$$

where $D$ is the maximum transportation distance in $R \cap \hat{R}$. Intuitively, an excursion can start from $x^s$ after time $r$ or from $x^d$ after time $r + d(x^s, x^d)$ to serve $R'$, whatever is the shorter of the two.

**An improved algorithm for** OLTSP **on the half-line metric** When restricting the metric space to $X = \mathbb{R}_{\geq 0}$, the best possible online algorithm is $\frac{3}{2}$-competitive [18]. We design a learning-augmented algorithm tailored to this metric space and a minimalistic prediction, namely, a single value $\hat{C}$ predicting the optimal makespan $C^*$. We prove the following error-dependent performance bound, which gives an almost tight consistency-robustness tradeoff w.r.t Theorem 5.

**Theorem 9.** *There is a learning-augmented algorithm for the half-line metric that has for every* $\alpha \in (0, 1/2]$ *a competitive ratio of at most* $\min \left\{ (1 + \alpha) \left( 1 + \frac{\Lambda_1}{C^*} \right), \frac{3}{2\alpha} \right\}$.

## 4   Online network design problems with predictions

This section sketches the applicability of our new error measure for the online-list problems Steiner Tree, Steiner Forest and (capacitated) facility location. To prove new error-dependent performance bounds as stated in Theorem 3 and Theorem 4, we revisit the algorithms proposed by Azar et al. [12] and analyze it w.r.t. our cover error measure. The key is an appropriate, problem-specific definition of the cost of a hyperedge and the corresponding analysis. Recall that the cost of a hyperedge $R' \cup \{x'\}$ should (intuitively) be equal to the value of an optimal solution for serving a set of unexpected (predicted absent) requests $R'$ w.r.t. a predicted (actual) request $x'$. We define the cost functions:

**Steiner tree:** $\gamma^{\mathrm{ST}}(R', x') =$ *cost of an optimal Steiner tree for terminals $R'$ with root $x'$.*

**Steiner forest:** $\gamma^{\mathrm{SF}}(R', x') =$ *cost of an optimal Steiner forest for terminal pairs $R'$ when connecting via $x' = (s', t')$ is free.*

**Facility location:** $\gamma^{\mathrm{FL}}(R', x') =$ *cost for opening facility $x'$ and assigning clients $R'$ to it.*

Our main technical contribution for online network design problems are the proofs of Theorem 3 and Theorem 4. We now give some intuition on how these proofs work by considering the online Steiner tree problem, and defer details and results for this and the other problems to the full version [16].

On a high level, the algorithm by Azar et al. [12] for the online Steiner tree problem does the following. Each new request (terminal) is connected to the current solution greedily by buying edges on a shortest path to a vertex of the current tree. When the Greedy cost increased sufficiently (with thresholds following a doubling-strategy), the algorithm spends a certain budget (depending on the spent Greedy cost) on connecting as many future predicted requests as possible to the current solution.

The proof of Theorem 3 splits the execution of this algorithm into two parts, where the first part considers the time until all predicted requests are satisfied, and the second part the remaining execution. We then use a sub-result provided by Azar et al. [12] which roughly bounds the total cost of the first part by the optimal solutions of $R$ and $\hat{R}$, and the total cost of the second part by the cost of the algorithm for serving specific subsequences of the request sequence. To further bound the first part, we use the structure of a min-cost $\infty$-hyperedge cover of $\hat{R}$ to prove an upper bound of at most $\mathcal{O}(1) \cdot \mathrm{OPT} + \mathcal{O}(1) \cdot \Gamma_\infty(\hat{R}, R)$. For the second part, we consider the total cost the Greedy algorithm incurs for a hyperedge of a min-cost $k$-hyperedge cover of $R$, and conclude by the bounded hyperedge size and Greedy properties that this is at most $\mathcal{O}(\log k)$ times the hyperedge cost, yielding a total bound of $\mathcal{O}(1) \cdot \mathrm{OPT} + \mathcal{O}(\log k) \cdot \Gamma_k(R, \hat{R})$. Here we especially use the fact that in our chosen partition of the algorithm's execution, any predicted terminal $\hat{x}$ which covers actual requests must have already been served in the first part.

## 5   Experiments

We performed various empirical experiments on real-world OLTSP instances that demonstrate the benefits of using our algorithms over classic online algorithms. We consider the road network of Manhattan [19, 41] and compose 100 instances of 10 requests each based on taxi pickup requests from a dataset offered by the NYC Taxi & Limousine Commission.[1] We compare SMARTTRUST with the classic online algorithms REPLAN [7], IGNORE [7, 24, 44] and SMARTSTART [7]; all algorithms use efficient TSP heuristics. We report for every experiment and instance the empirical competitive ratio, i.e. the average ratio between the algorithms performance and the approximated value of the optimal makespan, as well as error bars that denote the 95% confidence interval over all instances.

---

[1] `https://www1.nyc.gov/site/tlc/about/tlc-trip-record-data.page`, downloaded 02/05/22

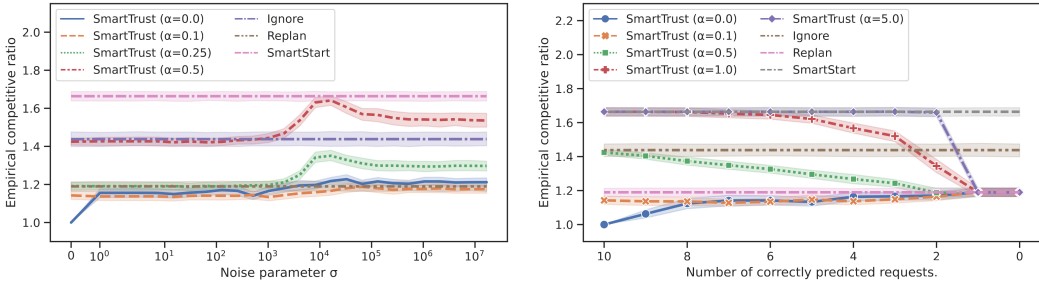

**(a)** Noise only in request locations        **(b)** Partial instance predicted correctly

**Figure 3:** Experimental results for two different prediction settings (100 instances with 10 requests each)

We sketch here two relevant experiments and defer further details to the full version [16]. The first experiment considers synthetic predictions with Gaussian noise $\sigma$ only in the request locations, i.e., the release dates are predicted correctly. The results (Figure 3a) show that SMARTTRUST with $\alpha = 0.1$ dominates classic algorithms even for arbitrarily bad predictions. In the second experiment only a part of the actual instance is predicted, which is an interesting and practice-relevant variant. Again, the results (Figure 3b) show that for small values of $\alpha$, SMARTTRUST outperforms all classic algorithms.

## Concluding remarks

The universal cover error can be applied to arbitrary problems with uncertain inputs. As it seems to be the first error measure that captures arrival times, it seems very natural to investigate, in particular, other online-time problems such as, e.g., scheduling problems. Further, it would be interesting to identify more compact or smaller predictions. While we predict a full input instance much less information might be sufficient to gain high-quality solutions. This can be only partial information about the input instance or predictions on algorithmic actions, such as an optimal tour instead of request sequences. In the latter case, we can directly apply our framework after approximating the predicted tour by some time-stamped discrete points and using those as input prediction.

## Acknowledgments and Disclosure of Funding

The work in this paper is supported in part by: the Netherlands Organisation for Scientific Research (NWO) through project OCENW.GROOT.2019.015 "Optimization for and with Machine Learning (OPTIMAL)" and Gravitation-grant NETWORKS-024.002.003; the German Science Foundation (DFG) under contract 146371743 – TRR 89 Invasive Computing; the ERC Advanced Grant 788893 AMDROMA "Algorithmic and Mechanism Design Research in Online Markets"; the MIUR - PRIN project ALGADIMAR "Algorithms, Games, and Digital Markets"; and the MUR - FSE REACT EU - PON R&I 2014-2020.

The map data used in experiments is copyrighted by OpenStreetMap contributors and available from https://www.openstreetmap.org.

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
