# OpenReview forum: "A Universal Error Measure for Input Predictions Applied to Online Graph Problems"
_NeurIPS.cc/2022/Conference — NeurIPS 2022 Accept_

### Official Review · Reviewer_xjD5 · 2022-07-06

**Rating:** 6
**Confidence:** 4
**Soundness:** 3 good
**Presentation:** 1 poor
**Contribution:** 3 good

**Summary:**

The paper studies learning-based online graph problems, focused on online TSP and its variants, online Steiner tree and forest, and online facility location. In each problem, there is a sequence of node requests arriving online, and an algorithm must move or select certain edges to serve these requests. Moving or buying edges come with costs, so the goal is to minimize the total cost, measured against the offline OPT.

Classic algorithms are focused on the worst-case competitive analysis and generally do not consider real world data distributions. In the algorithms with prediction model, the algorithm is instead provided with a predicted sequence of requests. We’d like to show that if the prediction is good, a learning-augmented algorithm can improve upon the classic worst-case guarantee. Otherwise, the algorithm should be robust and retain a worst-case competitive ratio.

The main contribution of the paper is a new error metric of the learned predictions (based on hypergraph edge cover). The paper claims that this metric bypasses some difficult cases of prior related work, and provides algorithms that work well given good predictions under this metric. Empirical evaluations demonstrate that the algorithms achieve good performance on real data.


**Questions:**


Algorithm 2 requires “predicted optimal cost C hat” as its input. I’d like to clarify that this is the OPT value of the predicted request sequence Rhat. Moreover, we know that optimally solving TSP is NP-hard. Can we solve C hat approximately up to a small constant? Based on the proof of lemma 6, I think we can actually provide a C hat that is, say, 3/2, approximation of the optimal Chat. This would worsen the final bound by only a fixed constant, though. Is that correct?


**Ethics Review Area:**

["I don’t know"]

**Limitations:**


Overall, I believe the paper is technically novel and has potential to make an impact. However, as it currently stands, it likely requires significant revision to improve its presentation. I have listed some concrete suggestions above.




**Strengths And Weaknesses:**


Strengths:
-----

The paper gives a novel notion of error metric of learned predictions in online graph problems.  Given the “dummy request example” (line 189 below), I think this new notion is reasonably motivated.  In particular, this example demonstrates a difficult case where the prediction is conceptually good, but not captured by the error metric of recent work, including Azar et al. [Online graph algorithms with predictions. SODA 22].

The technical claims in the full paper are sound (but I have not checked the proof details in the supplement).

The paper also gives experimental evaluations.


Weaknesses:
------

The paper is not well written, particularly the early sections. Some statements appear vague and only get specified (much) later, and sentences hard to parse. Below, I list a few for the author(s) to fix. I find they seriously obstruct my reading.

* Line 90: “Since we can use cost functions based…”  This sentence seems very vague to me. Is it a feature that prior work such as [12] doesn’t have? My understanding is that the matching-based error of [12] is also rather general and independent of the problem (but can be applied to all these node-arrival online graph problems).
* Line 92: “Further, it allows to integrate both actual and predicted release dates for online-time problems.” How? Up to this point, the cover error has not been fully specified, so I couldn’t tell why this is true.
* Line 97: “Although previously studied…” Regarding this dummy requests example, I suggest it’s written concretely right here in the intro. (I get that it’s defined later in line 189 and below)
* C* is undefined in Theorem 1.
* Line 131: “These bounds hold simultaneously for [...] the bounds provided in [12]” — What does this mean? (I get that it holds for any k, but not the second part.) Is it suggesting that these bounds recover some of the results in [12]?

Finally I suggest that the definition of the cover error be given earlier in the intro section.

Minor errors:
-----
Line 77: “In all our problems the cost of a hyperedge [...]”   In my first reading, this sentence feels long and a bit hard to parse. (What does “anchor” mean here?)

Line 80: “E.g….” This is not a complete sentence.

Line 85: “unavoidably” -> “unavoidable”

Line 89: I suggest using an itemize here to list these points

Line 90: “problem-independent” -> “problem-independently”

Figure 1: maybe state that the numbers on the nodes are just indices, and do not correspond to the order of the request sequence.

---

> ### Author Response · Authors · 2022-08-02
> **Response to Official Review of Paper9338 by Reviewer xjD5**
>
> ### Answers to questions:
>
> - Algorithm 2 requires 'predicted optimal cost $\hat{C}$': The reviewer correctly observed that Algorithm 2 makes this assumption; we have clarified it in the revision.
> - The reviewer also asks what happens if an approximate polynomial time algorithm is used instead of an optimal (exponential time) algorithm. We have considered the case of polynomial time algorithms in Section 3.2, that for space limitation is in Appendix B.2 in the supplementary material. Namely, Theorem 8 shows that by using  a $(1 + \nu)$-approximation algorithm for metric TSP to obtain this value and to recompute tours, and worsen the error-dependent ratio by a factor of at most $(1 + \nu)$. For the robustness bound we show a slightly better factor.
>
> ### Comments on weaknesses:
>
> The reviewer comments on the quality of the presentation particularly in the early sections. We regret that the reviewer finds the introductory part and the discussion on the results and impact too vague and we understand that he/she wishes precise definitions in the introduction instead of the clear technical part of the paper. We had chosen to address a more general public and keep the introduction in more general terms. The other reviewers appreciate this choice.  (One even raised the request to be less precise in the theorem statements.)
>
> We thank the reviewer for valuable comments that we have all implemented in the revision. Since the reviewer gave a positive evaluation of the proposed error function and of the technical contribution, we hope that he/she will decide to adjust the score given our changes.
>
> In the sequel we provide detailed comments of specific weaknesses
> - Line 90: The error definition of Azar et al. also uses arbitrary cost functions for the min-cost matching (although they do not choose a cost function based on the values of optimal solutions). The main advantage of our error is that we can also use asymmetric cost functions, allowing more general error definitions useful for more general/different problem types, e.g., for online-time problems we can capture lateness more precisely.
> - Line 90, 92 and 97: we have rewritten the text here to answer the reviewer's reservations.  Since it will be made precise later, it remains more of an announcement to the broad public than as a statement that the reader should see immediately. At this point, we just want to highlight the strengths of our error definition.
> - We added the definition of $C^*$ in the revision, and we also added what we mean with $\alpha$, $\Lambda_1$ and $\rho$.
> - Line 131: The second part notes that the bounds of Azar et al. still hold, which is of course trivial and we removed this in the revision. We still stress that the bounds of Azar et al. include a robustness bound of $O(\log |R|)$ for their algorithm.
>
> ### Comments on minor errors:
>
> Thank you for pointing us to these issues. We fixed all typos and addressed your suggestions in the revision.

---

### Official Review · Reviewer_Jom7 · 2022-07-11

**Rating:** 7
**Confidence:** 4
**Soundness:** 3 good
**Presentation:** 3 good
**Contribution:** 3 good

**Summary:**

This paper gives a new error measure for online algorithms with predictions for metric problems of two kinds: (a) online TSP and Dial-a Ride, and (b) online Steiner tree and forest. The two categories differ in their notion of online: the first category has requests released over (continuous) time, while the second category just has an online sequence of requests but no actual notion of time. The error measure is innovative and tries to address some shortcomings of previous measures. In particular, the paper compares to two papers from AAAI 22 and SODA 22. The first paper uses set differences to characterize error and the non-errorneous part of the prediction needs to exactly match the actual input locations. The second paper relaxes this notion by allowing the common set between predictions and actual input to not match exactly and uses a minimum cost matching between these points to quantify the quality of the match. At a high level, the deficiency being addressed in this paper is that this minimum cost matching (or the stricter requirement that the common part matches exactly) forces the cardinality of the common parts of the predicted and actual sets to be equal. So, for example, if there are many real input locations that are all close to a single predicted location, the previous error measures have large error while one can argue that the prediction is actually a good one in a qualitative sense. To address this, the new error measure being introduced in this paper (roughly speaking) replaces the edges in the matching with hyperedges with one point on one side (predicted or actual) and multiple points on the other side. The cost of the hyperedge is defined in a natural way based on the problem at hand.

Using this new measure, the paper gives the following results: (a) for online TSP and Dial-a-Ride, it gives online algorithms that degrade gracefully with the new notion of error, and (b) for Steiner tree and forest, it shows that the algorithm from the previous SODA 22 paper gracefully degrades with this notion of error. Although the algorithm in (b) is the one from the previous paper, the analysis with respect to the new error is new.

**Questions:**

1. Can you say more on the asymmetry of your error measure (using $\Lambda_\infty$ and $\Lambda_k$ in the two directions)? Is this necessary, i.e., does something go wrong if you defined the error measure symmetrically? This asymmetry actually shows up in other places as well (e.g., in the Almanza et al. work on online facility location with predictions where the bounds apply to the case when predictions contain all the actual requests but not vice-versa). So, I suspect this is an important (and necessary) asymmetry. The paper should include a discussion about this.

2. Theorems 1 and 2 should be replaced by informal statements if possible. At this point, $\Lambda_1$ hasn't even been defined formally, and as such, it is difficult to interpret the formal bounds of these theorems without knowing this definition precisely. I believe $C^*$ wasn't defined before the theorems either (but this is easier to fix by just adding the definition).

**Limitations:**

There is no explicit discussion of limitations. This can be included, perhaps in the conclusions section.

**Strengths And Weaknesses:**

Strengths:

1. I think the online algorithms with ML predictions (or more generally data-driven algorithms) area is a very important one. This paper is exactly in the direction that is particularly important for this area, which is to explore new paradigms and frameworks that are natural and have interesting technical challenges. As the paper correctly claims, there is no unanimity on what the correct measure of error is (as against fairly universal acceptance of the notions of consistency and robustness) and this is a significant handicap because each problem requires its own specialized techniques to showing smoothness bounds with error. So, I really like what the paper sets out to do, which is to give a reasonable notion of error that applies to many (at least metric) problems, and then give somewhat general techniques for obtaining smooth algorithms with respect to this error measure (at least techniques that extend beyond single problems).

2. The algorithms for online TSP and Dial-a-Ride with predictions are new. These are important problems in their own right, so initiating the study of online algorithms with predictions for these problems is an important step irrespective of the particular error measure being used here.

Weaknesses:

1. I did not completely understand why the paper eliminates the difference sets in the error measure that existed in the previous definitions. For instance, if the predictions are completely bogus, then it seems the actual location of the predictions should not matter. In particular, one should be able to recover robustness guarantees from error dependent bounds, but that does not seem to be the case here. I suspect it is possible to keep the current definition but only apply it to a common part (chosen in an adversarial manner) and allow difference sets outside the common part, thereby recovering robustness.

---

> ### Author Response · Authors · 2022-08-02
> **Response to Official Review of Paper9338 by Reviewer Jom7**
>
> ### Answer to question regarding asymmetry:
>
> Indeed this asymmetry is not necessary: since $\Gamma_{i+1} \leq \Gamma_{i}$, the error-dependent bounds also hold if we use hyperedges of size at most $k$ to cover actual requests (symmetric error $\Gamma_k(\hat{R}, R) + \Gamma_k(R, \hat{R})$ ). The reason why we allow arbitrarily sized hyperedges to cover predicted requests is that we noticed that for our problems we were always able to analyze the algorithms using $\Gamma_\infty$ to cover predicted requests, which gives tighter results. We included a comment of this aspect in the revision, after the definition of the cover error.
>
> ### Intuition on asymmetry:
>
> The intuitive reason why this works for our problems and why we believe that it will work for other problems is the following. In the analysis, the predicted requests are known in advance and they are independent of the algorithm. It is therefore easier to relate  the optimal solution of the predicted requests and the optimal solution of the actual requests in this full information setting, where we use this cover. This is opposed to the setting where we compare the cost of the online algorithm to the optimal solution of the predicted requests, where the online arrival of the actual requests makes it a lot harder to cover them by predicted requests.
>
> ### Answer to question regarding Theorems 1 and 2:
>
> We rephrased Theorem 1 and 2 as suggested. Even if not formulating algorithms precisely, we have added text above the theorems to give a meaning to the various concepts used in the theorem. We hope that this gives enough insight to read the theorems meaningfully, because we do prefer to keep the statements exact.
>
>
> ### Reply to the mentioned weakness:
>
> We are not sure whether we understand the question regarding "eliminat[ing] the difference sets in the error measure that existed in the previous definitions". Does it refer to error definitions using outliers as in Azar et al? For the class of online-list problems without constant competitive ratio (online Steiner tree, facility location, etc.) indeed it makes sense to use counting-based error measures to interpolate between constant and non-constant competitive ratios (not for TSP and DARP). We believe that we can use our min-cost cover error instead of a matching for a non-outlier set, which solves issues with the original matching-with-outliers definition.

---

### Official Review · Reviewer_UJPw · 2022-07-12

**Rating:** 6
**Confidence:** 3
**Soundness:** 3 good
**Presentation:** 3 good
**Contribution:** 3 good

**Summary:**

This paper introduces a new notion of error for algorithms with predictions in the context of online metric graph problems. The predictions considered are the set of all requests, e.g., the offline problem instance. At a high level, the "cover error" introduced in this paper measures the optimal solution cost on the set of requests that appear in the prediction but not in the real instance and vice versa. This is formalized as a min-cost bipartite hypergraph cover problem to account for these two types of error via "detours" from the true instance. The authors give instantiations of this general error measure to online TSP, online Steiner tree/forest, and online facility location and develop algorithms that use predictions and whose performance is parameterized by the cover error.

**Questions:**

Could the authors please elaborate on the weakness discussed above? In some sense, it seems like this paper brings the error to the algorithm as opposed to the more natural idea of designing an algorithm to work better with some standalone notion of error.

**Limitations:**

I think the discussion of limitations is adequate.

**Strengths And Weaknesses:**

Strengths
- The cover error has several benefits, which the authors do a good job highlighting. In particular, it accounts for the asymmetry in the impacts of the errors in which we predict requests that aren't there and errors in which we miss real requests in our predictions.
- The authors develop algorithms parameterized by the cover error for various online graph problems, achieving bounds that can significantly improve on the worst-case if the cover error is small.
- I think that this error measure and the resulting algorithms will be of interest to those working on algorithms with predictions.

Weaknesses
- Naturally, more complex error measures can more precisely capture the performance of algorithms with predictions. I worry that in some sense the definition of this error measure almost makes the fact that there exist good algorithm if this error is small a tautology. More basic error measures may be loose in certain respects, but they tell us something clear about the relation between the prediction and instance which we may or may not be able to leverage algorithmically: they have some meaning somewhat separate from the algorithmic task. It seems that saying the cover error is basically an algorithmic fact: there are a small set of errors which we can cover by work which we must do anyway.

---

> ### Author Response · Authors · 2022-08-02
> **Response to Official Review of Paper9338 by Reviewer UJPw**
>
> ### Answer to weaknesses:
>
> The cover error is more of a framework than a concrete error. It is also possible to express very basic problem-independent error measures in this framework. For example:
> - Only allow simple edges (k=1) and define a binary cost function based on whether a predicted point is correct or not, giving the total number of wrong predictions.
> - Only allow simple edges (k=1) and consider the metric distance between the two incident
> requests (similar to what we do to cover actual requests for online TSP).
> And at the same time, our framework allows more complex error measures tailored to specific problems such as we do in our paper (for larger k).
>
> In this new research area, there is currently no general agreement on what constitutes a good error measure. (This is in contrast to the general agreement on the notions of robustness and consistency.) Similar to the reviewer, we feel that there is the need for such a discussion. With our general error notion, we hope to contribute to it. Therefore we think that our work can be of high impact in this field.
>
> There is definitely a tradeoff between (a) a (rather) simple and problem-independent error measure which, in some cases, may essentially give no information on the quality of a prediction for solving a particular problem and (b) more  complex/problem-specific error measures which allow more precise measurements w.r.t. to a certain problem class.
>
> We believe that our framework can be used to steer this tradeoff through the cost functions and the parameter k. E.g., small hyperedges (small k) describe simple relations between predicted and actual requests, while large hyperedges (large k) can measure more complex, possibly problem-specific, dependencies between multiple predicted and actual requests. In any case, the community agrees that an error definition must be algorithm-independent. In this respect, notice that all our proposed cost functions are based on costs of optimal solutions for subinstances, independent of algorithmic costs and techniques.

---

> > ### Comment · Reviewer_UJPw · 2022-08-07
> > **Reply to authors**
> >
> > Thanks for your response! I agree with the big-picture that this is an interesting notion of error that addresses some gaps in the existing literature. I'm not entirely convinced that the cover error is the "right" notion of error in general but am sympathetic to the fact that this is very subjective and think that this work helps to further that discussion. In any case, I think that this paper makes an interesting contribution to algorithms with predictions.

---

### Official Review · Reviewer_6xM7 · 2022-07-12

**Rating:** 6
**Confidence:** 4
**Soundness:** 4 excellent
**Presentation:** 3 good
**Contribution:** 3 good

**Summary:**

The paper considers online algorithms for classic graph problems when predictions regarding the requests are provided to the algorithm. The key contribution of the paper is to define a new notion of error to measure the quality of predictions provided for such problems. The new error attempts to capture the intuition that if following the predictions leads to substantially worse performance than the optimal solution, then such predictions should suffer from large errors. To make this notion precise, the paper defines the error as the size of the minimum cost edge cover in an associated hypergraph where the edge costs capture the amount of excess cost required to satisfy incorrectly predicted requests.

For the online traveling salesman problem (and its dial-a-ride generalization), the paper provides a framework to “combine” a known robust online algorithm with an offline TSP algorithm on the predicted requests to obtain consistent and robust algorithms. As in many prior work, the algorithm is parameterized by an alpha that represents the trust of the decision maker in the predictions and simultaneously provides a competitive ratio of (1+alpha) if the predictions are correct, and also O(\rho/\alpha) if the predictions are incorrect where \rho is the competitiveness of the base algorithm.

Using the new error measure, the paper also reanalyzes the algorithm by Azar et al for online steiner tree / forest with predictions and show that it obtains a total cost of at most O(1) OPT + f(k, error) [as opposed to the logarithmic dependence on the error in the multiplier for OPT in Azar et al.] yielding tighter guarantees for certain predictions.


**Questions:**

1. In the definition of Cover Error in section 2, it is not mentioned how the hyperedges are constructed. Does the hypergraph G_k have all possible hyperedges of the specified size?
2. Related to the point above, since the hyperedges (and their costs) are not defined yet, the statement that \Gamma_i((A,B) >= \Gamma_{i+1}(A,B) [on line 179] is not well justified.


**Strengths And Weaknesses:**

The paper is well motivated and the new error formulation is pretty intuitive for graph problems.
Overall I quite like the paper and find the contributions to be non-trivial and interesting. The paper is well-written and reads well. The experimental section is also well written and the experiments are pretty comprehensive for a primarily theoretical paper.

---

> ### Author Response · Authors · 2022-08-02
> **Response to Official Review of Paper9338 by Reviewer 6xM7**
>
> ### Answers to questions:
>
> 1. The sentence in Lines 171 and 172 should describe that we consider all hyperedges of the specified size or smaller, as the reviewer also concluded. We fixed this definition in the revision by explicitly describing the construction of the set H.
> 2. Any i-hyperedge cover is also an (i+1)-hyperedge cover. Therefore the cost of the minimum-cost (i+1)-hyperedge cover is at most the cost of the minimum-cost i-hyperedge cover. This holds for any hyperedge cost function.

---

### Meta-Review · Area_Chair_g4as · 2022-08-27

**Recommendation:** Accept
**Confidence:** Certain

**Metareview:**

"Algorithms via (ML-based) predictions"---especially for online problems---is a young, fast-growing, important area. Of course, the predictions will usually not be perfect and will involve some sort of error. As this area is nascent, it is vital to develop and analyze different forms of error and for various fundamental problems, which this paper does well. In particular, this work develops a new notion of error for two types of "metric" problems in the above genre: online TSP and Dial-a Ride, and online Steiner tree/forest. The first type has arrivals over continuous time, while the second has an online request-sequence (as is typical in the algorithmic study of online problems). The error measure addresses some shortcomings of previous measures, and compares to recent works from Xu et al. (AAAI '22) and Azar et al. (SODA '22). Xu et al. use set-differences to characterize error; the non-erroneous part of the prediction has to exactly match the input locations. Azar et al. relax this by allowing the common set between predictions and actual input to not match exactly and use the cost of a min-cost matching between these points to quantify the extent of the match. The gap addressed in the present paper is that these two types of works force the cardinality of the common parts of the predicted and actual sets to be equal. The present paper's error measure essentially replaces the matching with hyperedges e with one vertex on one side (predicted or actual) and multiple vertices on the other side; cost of e is defined based on the problem.

This paper develops the following results parametrized by this error: online TSP and Dial-a-Ride---online algorithms that degrade gracefully with the error, and Steiner tree/forest---showing that the algorithm of Azar et al. does indeed gracefully degrades with this error.

The paper was generally appreciated by the reviewers; the authors are encouraged to take the review comments into account.

**Award:**

No

---

### Decision · Program_Chairs · 2022-09-14

Accept